# *MagicDrive3D*: Controllable 3D Generation for Any-View Rendering in Street Scenes

## Abstract

While controllable generative models for images and videos have achieved remarkable success, high-quality models for 3D scenes, particularly in unbounded scenarios like autonomous driving, remain underdeveloped due to high data acquisition costs. In this paper, we introduce *MagicDrive3D*, a novel pipeline for controllable 3D street scene generation that supports multi-condition control, including BEV maps, 3D objects, and text descriptions. Unlike previous methods that reconstruct before training the generative models, *MagicDrive3D* first trains a video generation model and then reconstructs from the generated data. This innovative approach enables easily controllable generation and static scene acquisition, resulting in high-quality scene reconstruction. To address the minor errors in generated content, we propose deformable Gaussian splatting with monocular depth initialization and appearance modeling to manage exposure discrepancies across viewpoints. Validated on the nuScenes dataset, *MagicDrive3D* generates diverse, high-quality 3D driving scenes that support any-view rendering and enhance downstream tasks like BEV segmentation. Our results demonstrate the framework's superior performance, showcasing its transformative potential for autonomous driving simulation and beyond.

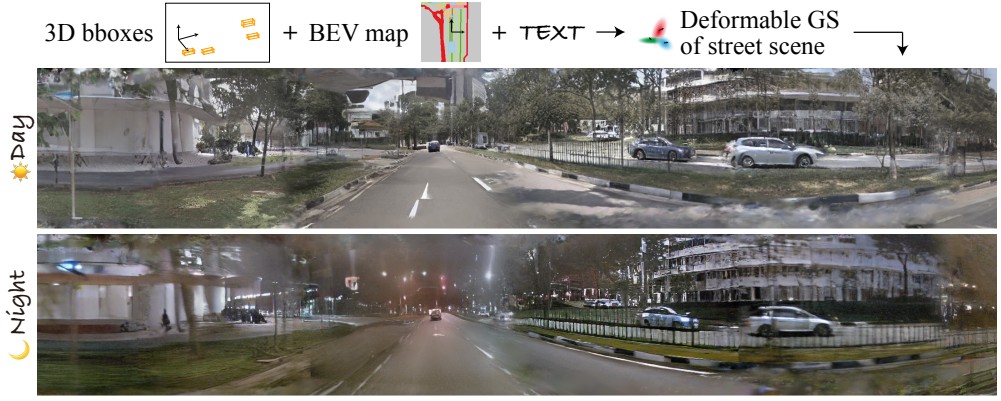

Figure 1: Rendered panorama of the street scene generated from *MagicDrive3D*. With conditional controls from 3D bounding boxes of objects, BEV road map, and text descriptions (*e.g.*, weather), *MagicDrive3D* generates complex open-world 3D scenes represented by deformable Gaussians.

## 1 Introduction

With the advancement of generative models, particularly diffusion models (Goodfellow et al., 2014; Ho et al., 2020; Song et al., 2020; Rombach et al., 2022), there has been increasing interest in generating 3D assets (Poole et al., 2022; Vahdat et al., 2022; Bautista et al., 2022). While a significant amount of work has focused on object-centric generation (Poole et al., 2022; Tang et al., 2024), generating open-ended 3D scenes remains relatively unexplored. This gap is even more critical because

---

Project Page: https://magicdrive3d.github.io/.

many downstream applications, such as Virtual Reality (VR) and autonomous driving simulation, require controllable generation of 3D street scenes, which is an open challenge.

3D-aware view synthesis[1] methods can be broadly categorized into two approaches: geometry-free view synthesis and geometry-focused scene generation (Rombach et al., 2021). Geometry-free methods directly generate 2D images (Chen et al., 2024) or videos (Gao et al., 2024; Wen et al., 2023; Wang et al., 2023b) based on camera parameters, excelling in photo-realistic image generation. However, they often lack sufficient geometric consistency, limiting their ability to extend to viewpoints beyond the dataset (Gao et al., 2024; Wen et al., 2023; Wang et al., 2023b). On the other hand, geometry-focused methods (*e.g.*, GAUDI (Bautista et al., 2022) and NF-LDM (Kim et al., 2023)) generate 3D representations (*e.g.*, NeRF (Mildenhall et al., 2020) or voxel grids) from latent inputs, supporting multi-view rendering. Despite their broader applicability, these methods require expensive data collection, necessitating static scenes and consistent sensor properties like exposure and white balance. Street view datasets, such as nuScenes (Caesar et al., 2020), often fail to meet these requirements, making it extremely challenging to train geometry-focused 3D street scene generation models using such datasets.

Recognizing the advancements in controllable generation by geometry-free view synthesis methods (Chen et al., 2024; Gao et al., 2024), it is potential to use them as data engines. Their controllability and photo-realism could address the challenges faced by geometry-focused methods. However, the limited 3D consistency in synthetic views from geometry-free methods, such as temporal inconsistency among frames and deformation of objects, poses crucial issues for integrating two kinds of methods into a unified framework.

To address these challenges, we propose *MagicDrive3D*, a novel framework that combines geometry-free view synthesis and geometry-focused reconstruction for controllable 3D street scene generation. As illustrated in Figure 2, our approach begins with training a multi-view video generation model to synthesize multiple views of a static scene. This model is configured using controls from object boxes, road maps, text prompts, and camera poses. To enhance inter-frame 3D consistency, we incorporate coordinate embeddings that represent the relative transformation between LiDAR coordinates for accurate control of frame positions. Next, we improve the reconstruction quality of generated views by enhancing 3D Gaussian splatting from the perspectives of prior knowledge, modeling, and loss functions. Given the limited overlap between different camera views (Xie et al., 2023), we adopt a monocular depth prior and propose a dedicated algorithm for alignment in sparse-view settings. Additionally, we introduce deformable Gaussian splatting and appearance embedding maps to handle local dynamics and exposure discrepancies, respectively.

Demonstrated by extensive experiments, our *MagicDrive3D* framework excels in generating highly realistic street scenes that align with road maps, 3D bounding boxes, and text descriptions, as exemplified in Figure 1. We show that the generated camera views can augment training for Bird's Eye View (BEV) segmentation tasks, providing comprehensive controls for scene generation and enabling the creation of novel street scenes for autonomous driving simulation. Notably, *MagicDrive3D* is the first to achieve controllable 3D street scene generation using a common driving dataset (e.g., the nuScenes dataset (Caesar et al., 2020)), without requiring repeated data collection from static scenes.

We summarize our contributions as follows:

- We propose *MagicDrive3D*, the first framework to effectively integrate both geometry-free and geometry-focused view synthesis for controllable 3D street scene generation. *MagicDrive3D* generates realistic 3D street scenes that support rendering from any camera view according to various control signals.

- We introduce a relative pose embedding technique to generate videos with improved 3D consistency. Additionally, we enhance the reconstruction quality with tailored techniques, including deformable Gaussian splatting, to handle local dynamics and exposure discrepancies in the generated videos.

- Through extensive experiments, we demonstrate that *MagicDrive3D* generates high-quality street scenes with multi-dimensional controllability. Our results also show that synthetic data improves 3D perception tasks, highlighting the practical benefits of our approach.

---

[1]In this paper, we focus on generative models where views/scenes are generated from latent variables.

## 2 RELATED WORK

**3D Scene Generation**. Numerous 3D-aware generative models can synthesize images with explicit camera pose control (Zhao et al., 2024; Rombach et al., 2021) and potentially other scene properties (Tang et al., 2023), but only a few scale for open-ended 3D scene generation. GSN (DeVries et al., 2021) and GAUDI (Bautista et al., 2022), representative of models generating indoor scenes, utilize NeRF (Mildenhall et al., 2020) with latent code input for "floorplan" or tri-plane feature. Their reliance on datasets covering different camera poses is incompatible with typical driving datasets where camera configuration remains constant. NF-LDM (Kim et al., 2023) develops a hierarchical latent diffusion model for scene feature voxel grid generation. However, their representation and complex modeling hinder fine detail generation.

Contrary to previous works focusing on scene generation using explicit geometry, often requiring substantial data not suitable for typical street view datasets (*e.g.*, nuScenes (Caesar et al., 2020)) as discussed in Section 1, we propose merging geometry-free view synthesis with geometry-focused scene representations for controllable street scene creation. Methodologically, LucidDreamer (Chung et al., 2023) is most similar to our approach, although it relies on a text-controlled image generation model, which cannot qualify as a view synthesis model. In contrast, our video generation model is 3D-aware. Besides, we propose several improvements over 3DGS for better scene generation quality. Besides, inpainting-based methods like LucidDreamer (Chung et al., 2023) and WonderJourney (Yu et al., 2024) cannot complete our controllable street scene generation task. We showcase the differences in Appendix C.

**Street View Video Generation**. Diffusion models (Song et al., 2020; Ho et al., 2020) have influenced a range of works on street view video generation, from single to multi-view videos (*e.g.*, (Wang et al., 2023a; Gao et al., 2024; Wen et al., 2023; Wang et al., 2023b)). Despite cross-view consistency being essential for multi-view video generation, their generalization ability on camera poses is limited due to their data-centric nature (Gao et al., 2024). Furthermore, these models lack exact control over frame transformation (*i.e.*, precise car trajectory), which is crucial for scene reconstruction. Our work addresses this by enhancing control in video generation and proposing a dedicated deformable Gaussian splatting for geometric assurance.

**Street Scene Reconstruction**. Scene reconstruction and novel view rendering for street views are useful in applications like driving simulation, data generation, and augmented and virtual reality. For street scenes, attributes like scene dynamic and discrepancies from multi-camera data collection make typical large-scale reconstruction methods ineffective (*e.g.*, Rematas et al. (2022); Martin-Brualla et al. (2021); Lin et al. (2024)). Hence, real data-based reconstruction methods like Xie et al. (2023); Yan et al. (2024) utilize LiDAR for depth prior, but their output only permits novel view rendering from the same scene. Unlike these methods, our approach synthesizes novel scenes under multiple levels of conditional controls.

## 3 PRELIMINARIES

**Problem Formulation**. In this paper, we focus on controllable street scene generation. Given scene description $\mathbf{S}_t$, our task is to generate street scenes (represented with 3D Gaussians $\mathbf{G}$) that correspond to the description from a set of latent $\mathbf{z} \sim \mathcal{N}(\mathbf{0}, \mathbf{I})$, *i.e.* $\mathbf{G} = \mathcal{G}(\mathbf{S}_t, \mathbf{z})$. To describe a street scene, we adopt the most commonly used conditions as per Gao et al. (2024); Wang et al. (2023b); Wen et al. (2023). Specifically, a frame of driving scene $\mathbf{S}_t = \{\mathbf{M}_t, \mathbf{B}_t, \mathbf{L}_t\}$ is described by road map $\mathbf{M}_t \in \{0, 1\}^{w \times h \times c}$ (a binary map representing a $w \times h$ meter road area in BEV with $c$ semantic classes), 3D bounding boxes $\mathbf{B}_t = \{(c_i, b_i)\}_{i=1}^N$ (each object is described by box $b_i = \{(x_j, y_j, z_j)\}_{j=1}^8 \in \mathbb{R}^{8 \times 3}$ and class $c_i \in \mathcal{C}$), and text $\mathbf{L}_t$ describing additional information about the scene (*e.g.*, weather and time of day). In this paper, we parameterize all geometric information according to the LiDAR coordinate of the ego car.

One direct application of scene generation is any-view rendering. Specifically, given any camera pose $\mathbf{P} = [\mathbf{K}, \mathbf{R}, \mathbf{t}]$ (*i.e.*, intrinsics, rotation, and translation), the model $\mathbf{G}(\cdot)$ should render photo-realistic views with 3D consistency, $\mathcal{I}^r = \mathbf{G}(\mathbf{P})$, which is not applicable to previous street view generation (*e.g.*, Gao et al. (2024); Wang et al. (2023b); Wen et al. (2023)). Besides, we present more applications in Section 5.

Figure 2: Method Overview of *MagicDrive3D*. For controllable street scene generation, *Magic-Drive3D* decomposes the task into two steps: ① conditional multi-view video generation, which tackles the control signals and provides detailed prior of the scene; and ② scene reconstruction with deformable Gaussian splatting, which guarantees view consistency for any-view rendering.

**3D Gaussian Splatting**. We briefly introduce 3DGS since our scene representation is based on it. 3DGS (Kerbl et al., 2023) represents the geometry and appearance via a set of 3D Gaussians **G**. Each 3D Gaussian is characterized by its position $\boldsymbol{\mu}_p$, anisotropic covariance $\boldsymbol{\Sigma}_p$, opacity $\alpha_p$, and spherical harmonic coefficients for view-dependent colors $c_p$. Given a sparse point cloud $\mathcal{P}$ and several camera views $\{\mathcal{I}_i\}$ with poses $\{\mathbf{P}_i\}$, a point-based volume rendering (Zwicker et al., 2001) is applied to make Gaussians optimizable through gradient descend and interleaved point densification. Specifically, the loss is as follows:

$$\mathcal{L}_{\text{GS}} = (1 - \lambda)\mathcal{L}_1(\mathcal{I}_i^r, \mathcal{I}_i) + \lambda\mathcal{L}_{\text{D-SSIM}}(\mathcal{I}_i^r, \mathcal{I}_i), \tag{1}$$

where $\mathcal{I}^r$ is the rendered image, $\lambda$ is a hyper-parameter, and $\mathcal{L}_{\text{D-SSIM}}$ denotes the D-SSIM loss (Kerbl et al., 2023).

## 4 METHODS

In this section, we introduce our controllable street scene generation pipeline. Due to the challenges that exist in data collection, we integrate geometry-free view synthesis and geometry-focused reconstruction, and propose a generation-reconstruction pipeline, detailed in Section 4.1 and Figure 2. Specifically, we introduce a controllable video generation model to connect control signals with camera views (Section 4.2) and enhance the 3DGS from prior, modeling and loss perspectives (Section 4.3) for better reconstruction with generated views.

### 4.1 3D STREET SCENE GENERATION

Direct modeling of controllable street scene generation faces two major challenges: scene dynamics and discrepancy in data collection. *Scene dynamics* refer to the movements and deformation of elements in the scene, while *discrepancy in data collection* refer to the discrepancy (*e.g.*, exposure) caused by data collection. These two challenges are even more severe due to the sparsity of cameras for street views (*e.g.*, typically only 6 surrounding perspective cameras). Therefore, reconstruction-generation frameworks do not work well for street scene generation (Kim et al., 2023; Bautista et al., 2022).

Figure 2 shows the overview of *MagicDrive3D*. Given scene descriptions **S** as input, *MagicDrive3D* first extend the descriptions into sequence $\{\mathbf{S}_t\}$, where $t \in [0, T]$ according to preset camera poses $\{\mathbf{P}_{c,t}\}$, and generate a sequence of successive multi-view images $\{\mathcal{I}_{c,t}\}$, where $c \in \{1, \ldots, N\}$ refers to $N$ surrounding cameras, according to conditions $\{\mathbf{S}_t, \mathbf{P}_{c,t}\}$ (detailed in Section 4.2). Then we construct Gaussian representation of the scene with $\{\mathcal{I}_{c,t}\}$ and camera poses $\{\mathbf{P}_{c,t}\}$ as input. This step contains an initializing procedure with a pre-trained monocular depth model and an optimizing process with deformable Gaussian splatting (detailed in Section 4.3). Consequently, the generated street scene not only supports any-view rendering, but also accurately reflects different control signals.

*MagicDrive3D* integrate geometry-free view synthesis and geometry-focused reconstruction, where control signals are tackled by a multi-view video generator, while reconstruction step guarantee the generalization ability for any-view rendering. Such a video generator has two advantages: first, since multi-view video generation does not require generalization on novel views (Gao et al., 2024),

it poses less data dependency for street scenes; second, through conditional training, the model is capable of decomposition of control signals, and thus turns dynamic scenes into static scenes which are more friendly for reconstruction. Besides, for the reconstruction step, strong prior from the multi-view video reduces the burden for scene modeling with complex details.

## 4.2 RELATIVE POSE CONTROL FOR VIDEO GENERATION

Given scene descriptions and a sequence of camera poses $\{\mathbf{S}_t, \mathbf{P}_{c,t}\}$, our video generator is responsible for multi-view video generation. Although many previous art for street view generation achieve expressive visual effects, such as Gao et al. (2024); Wen et al. (2023); Wang et al. (2023b;a), their formulations leave out a crucial requirement for 3D modeling. Specifically, the camera pose $\mathbf{P}_{c,t}$ is typically relative to the LiDAR coordinate of each frame. Thus, there is no precise control signal related to the ego trajectory, which significantly determine the geometric relationship between views of different $t$s.

In our video generation model, we amend such precise control ability by adding the transformation between each frame to the first frame, *i.e.*, $\mathbf{T}_t^0$. To properly encode such information, we adopt Fourier embedding with Multi-Layer Perception (MLP), and concatenate the embedding with the original embedding of $\mathbf{P}_{c,t}$, similar to Gao et al. (2024). As a result, our video generator provides better 3D consistency across frames, most importantly, making the camera poses to each view available in the same coordinate, *i.e.*, $[\mathbf{R}_{c,t}^0, \mathbf{t}_{c,t}^0] = \mathbf{T}_t^0[\mathbf{R}_{c,t}, \mathbf{t}_{c,t}]$.

## 4.3 ENHANCED GAUSSIAN SPLATTING FOR GENERATED CONTENT

As introduced in Section 3, 3DGS is a flexible explicit representation for scene reconstruction. Besides, the fast training and rendering speed of 3DGS make it highly suitable for reducing generation costs in our scene creation pipeline. However, similar to other 3D reconstruction methods, 3DGS necessitates high cross-view 3D consistency at the pixel level, which unavoidably magnifies the minute errors in the generated data into conspicuous artifacts. Therefore, we propose improvements for 3DGS from the perspectives of *prior*, *modeling*, and *loss*, enabling 3DGS to tolerate minor errors in the generated camera view, thereby becoming a potent tool for enhancing geometric consistency in rendering.

**Prior: Consistent Depth Prior**. As essential geometry information, depth is extensively utilized in street scene reconstruction, such as the depth value from LiDAR or other depth sensors used in Xie et al. (2023); Yan et al. (2024). However, for synthesized camera views, the depth is unavailable. Therefore, we propose to use pre-trained monocular depth estimator (Bhat et al., 2023) to infer depth information.

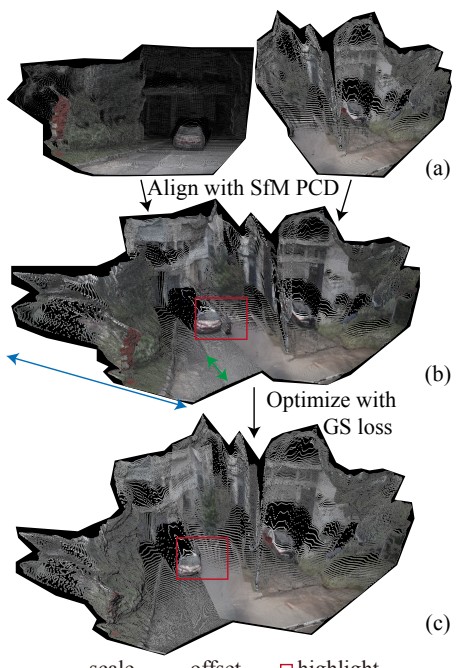

Figure 3: We optimize the monocular depths (a) with 2 steps for better alignment: coarse scale/offset estimation with SfM PCD (b) and GS optimization (c).

While monocular depth estimation is separate for each camera view, proper scale $s_{c,t}$ and offset $b_{c,t}$ parameters should be estimated to align them for a single scene (Zhou et al., 2023), as in Figure 3(a). To this end, we first apply the Point Cloud (PCD) from Structure of Motion (SfM) (Schönberger et al., 2016; Schönberger & Frahm, 2016), shown in Figure 3(b). However, such PCD is too sparse to accurately restore $(s_{c,t}, b_{c,t})$ for any views. To bridge the final gap, secondly, we propose further optimizing the $(s_{c,t}, b_{c,t})$ using the GS loss, as in Figure 3(c). Specifically, we replace the optimization for Guassian centers $\boldsymbol{\mu}_i$ with $(s_{c,t}, b_{c,t})$. After the optimization, we initialize $\boldsymbol{\mu}_i$ with points from depth values. Since GS algorithm is sensitive to accurate point initialization (Kerbl et al., 2023; Fan et al., 2024), our method provides useful prior to reconstructing in this sparse view scenario.

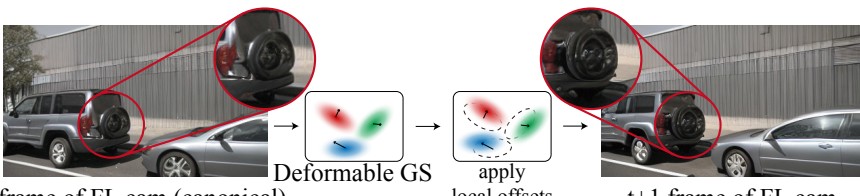

Figure 4: Illustration of the local dynamic from two successive generated frames of Front-Left (FL) camera. Even though our video generation model retains fine 3D consistency, minor discrepancies are inevitable. Our DGS can effectively reconstruct the scene with awareness of such discrepancy.

**Modeling: Deformable Gaussians for Local Dynamic**. Despite the 3D geometric consistency provided by our video generation model, there are inevitably pixel-level disparities in some object details, as shown in Figure 4. The strict consistency assumption of 3DGS may amplify these minor errors, resulting in floater artifacts. To mitigate the impact of these errors, we propose Deformable Gaussian Splitting (DGS), which, based on 3DGS, reduces the requirement for temporal consistency between frames, thereby ensuring the reconstruction effect of the generated viewpoint.

Specifically, as shown in Figure 4, we pick the center frame $t = t_C$ as the canonical space and enforce all Gaussians in this space. Hence, we allocate a set of offsets to each Gaussian, $\boldsymbol{\mu}_p^o(t) \in \mathbb{R}^3$, where $t \in [1, \ldots, T]$ and $\boldsymbol{\mu}_p^o(t_C) \equiv \mathbf{0}$. Note that, different camera views from the same $t$ share the same $\boldsymbol{\mu}_p^o(t)$ for each Gaussian, and we apply regularization on them to keep the dynamic in local, as shown in Equation 2:

$$\mathcal{L}_{\text{reg}_o} = \|\boldsymbol{\mu}^o(t)\|_2. \tag{2}$$

Consequently, $\boldsymbol{\mu}_p^o(t)$ can manage the local dynamics driven by pixel-level disparities, while $\boldsymbol{\mu}$ focuses on the global geometric correlations. It ensures the quality of scene reconstruction by leveraging consistent parts across different viewpoints, simultaneously eliminating artifacts. Besides, with the analytical gradient w.r.t. $SE(3)$ pose of cameras (Matsuki et al., 2024), we also make the camera pose optimizable in the final few steps of GS iterations, which helps to mitigate the local dynamic from camera poses.

**Loss: Aligning Exposure with Appearance Modeling**. Typical street view dataset is collected with multiple cameras, which capture views independently through auto-exposure and auto-white-balance (Caesar et al., 2020). Since the video generation is optimized to match the original data distribution, the differences from different cameras also exist in the generated data. The appearance differences are well-known issues for in-the-wild reconstruction (Martin-Brualla et al., 2021). In this paper, we propose a dedicated appearance modeling technique for GS representation.

We hypothesize that the disparity between different views can be represented by affine transformations $\mathbf{A}_i(\cdot)$ for $i$-th camera view. An Appearance Embedding (AE) map $\mathbf{e}_i \in \mathbb{R}^{w_e \times h_e \times c_e}$ is allocated for each view, and a Convolutional Neural Network (CNN) is utilized to approximate this transformation matrix $w_{\mathbf{A}} \in \mathbb{R}^{w \times h \times 3}$ (Appendix D contains more details). The final computation of the pixel-wise $\ell_1$ loss is conducted using the transformed image. Therefore, our final loss for DGS is as follows:

---

**Algorithm 1** Enhanced Deformable GS
**Input:** camera views $\{\mathcal{I}_i\}$, camera parameters $\{\mathbf{P}_i^0\}$, monocular depth $\{\mathcal{D}_i\}$, optimization steps for depth $s_D$, camera pose $s_C$, and GS $s_{\text{GS}}$
**Output:** DGS of the scene $\{\boldsymbol{\mu}_p, \boldsymbol{\mu}_p^o, \boldsymbol{\Sigma}_p, \text{SH}_p\}$, and optimized camera pose $\{\mathbf{P}_i^0\}$
1: $\mathcal{P}_{SfM}$ = PCD from SfM
2: Optimize $(s_{c,t}, b_{c,t})$ with $\mathcal{P}_{\text{SfM}}$ for each $\{c, t\}$
3: Random initialize AEs $\{\mathbf{e}_i\}$
4: **for** step in $1, \ldots, s_D$ **do**
5:     Random pick one view $\mathcal{I}_i$
6:     $\mathcal{L} = \mathcal{L}_{\text{AEGS}}(\mathcal{I}_i, \mathcal{I}_i^r, \mathbf{e}_i)$
7:     Update $(s, b), \mathbf{e}_i, \boldsymbol{\Sigma}, \text{SH}$ with $\nabla\mathcal{L}$
8: **end for**
9: Initialize $\boldsymbol{\mu}$ with $(s, b)$ and $\mathcal{D}$
10: **for** step in $s_D, \ldots, s_{\text{GS}}$ **do**
11:     Random pick one view $\mathcal{I}_i$ and get its $t$
12:     $\mathcal{L} = \mathcal{L}_{\text{DGS}}(\mathcal{I}_i, \mathcal{I}_i^r, \mathbf{e}_i, \boldsymbol{\mu}^o(t))$
13:     Update $\boldsymbol{\mu}, \boldsymbol{\mu}^o(t), \mathbf{e}_i, \boldsymbol{\Sigma}, \text{SH}$ with $\nabla\mathcal{L}$
14:     **if** step $> s_C$ **then**
15:         Update $\mathbf{P}_i^0$ with $\nabla\mathcal{L}$
16:     **end if**
17: **end for**

---

$$\mathcal{L}_{\text{DGS}} = \mathcal{L}_{\text{AEGS}} + \lambda_{\text{reg}_o}\mathcal{L}_{\text{reg}_o} = (1 - \lambda)\mathcal{L}_1(\mathbf{A}_i(\mathcal{I}_i^r), \mathcal{I}_i) + \lambda\mathcal{L}_{\text{D-SSIM}}(\mathcal{I}_i^r, \mathcal{I}_i) + \lambda_{\text{reg}_o}\mathcal{L}_{\text{reg}_o}, \tag{3}$$

where $\lambda_{\text{reg}_o}$ is the hyper-parameter for offset regularization.

**Optimization Flow**. We demonstrate the overall optimization flow of the proposed DGS in Algorithm 1. Line 2 is the first optimization of monocular depths. Lines 4-8 refer to the second opti-

NeuralField-LDM (w/ in-house dataset)

Original 3DGS (w/ nuScens dataset)

Ours (w/ nuScens dataset)

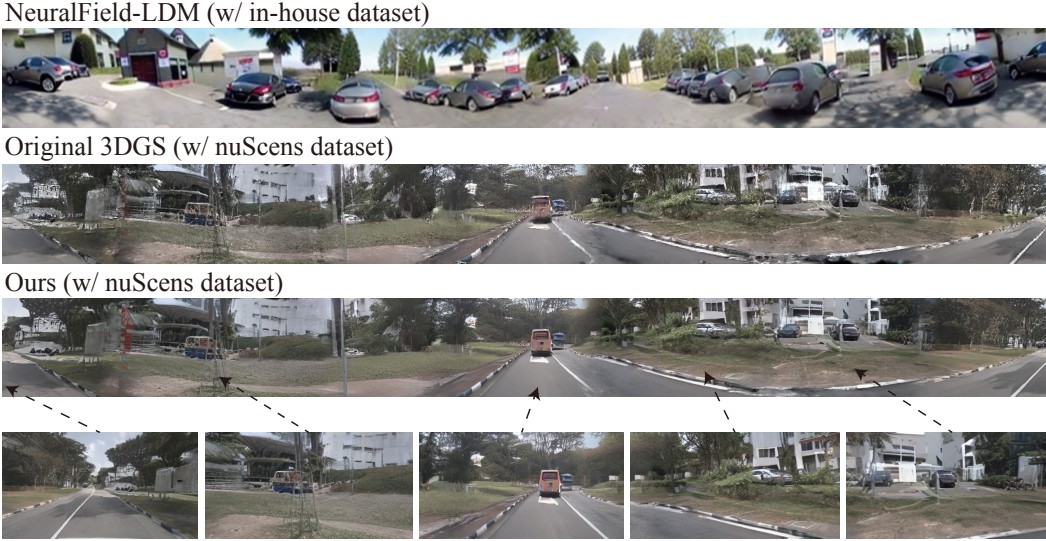

Figure 5: Qualitative comparison with NF-LDM (figure from Kim et al. (2023)) and original 3DGS (on the same generated video). Panoramas for GS are transformed and stitched from perspective cameras with 90° FOV. Views in the last row are rendered with camera rigs different from the nuScenes dataset.

mization of the monocular depths. Lines 10-16 are the main loop for DGS reconstruction, where we consider temporal offsets on Gaussians, camera pose optimization for local dynamics, and AEs for appearance discrepancies among views.

# 5 EXPERIMENTS

## 5.1 EXPERIMENTAL SETUP

**Dataset**. We test our *MagicDrive3D* using the nuScenes dataset (Caesar et al., 2020), which is commonly used for generating and reconstructing street views (Gao et al., 2024; Wen et al., 2023; Wang et al., 2023b; Xie et al., 2023). The official configuration is followed, using 700 street-view video clips of approximately 20s each for training and another 150 clips for validation. For semantics in control signals, we follow Gao et al. (2024), using 10 object classes and 8 road classes.

**Metrics and Settings**. *MagicDrive3D* is primarily evaluated using the Fréchet Inception Distance (FID) by rendering novel views unseen in the dataset and comparing their FID with real images. In addition, the method's video generation ability is evaluated using Fréchet

Table 1: Two settings for reconstruction quality evaluation. Testing views are in green while training views are in red.

| name | #test | #train | camera poses |
|------|-------|--------|--------------|
| 360° | 6 | 90 | |
| vary-t | 12 | 84 | |

Video Distance (FVD), and its reconstruction performance is assessed using L1, PSNR, SSIM (Wang et al., 2004), and LPIPS (Zhang et al., 2018). For reconstruction evaluation, two testing scenarios are employed: 360°, where all six views from $t = 9$ are reserved for testing the reconstruction in the canonical space; and vary-t, where one view is randomly sampled from different $t$ to assess long-range reconstruction ability through $t$ in the canonical space (as shown in Table 1).

**Implementation**. For video generation, we train our generator based on the pre-trained street view image generation model from Gao et al. (2024). By adding the proposed relative pose control, we train 4 epochs (77040 steps) on the nuScenes training set with a learning rate of $8e^{-5}$. We follow the settings for 7-frame videos described in Gao et al. (2024), using $224 \times 400$ for each view but extending to $T = 16$ frames. Consequently, for reconstruction, we select $t = 8$ as the canonical space. Except we change the first 500 steps to optimize $(s_{c,t}, b_{c,t})$ for each view and $\lambda_{\text{reg}_o} = 1.0$, other settings are the same as 3DGS. More details can be found in Appendix A.

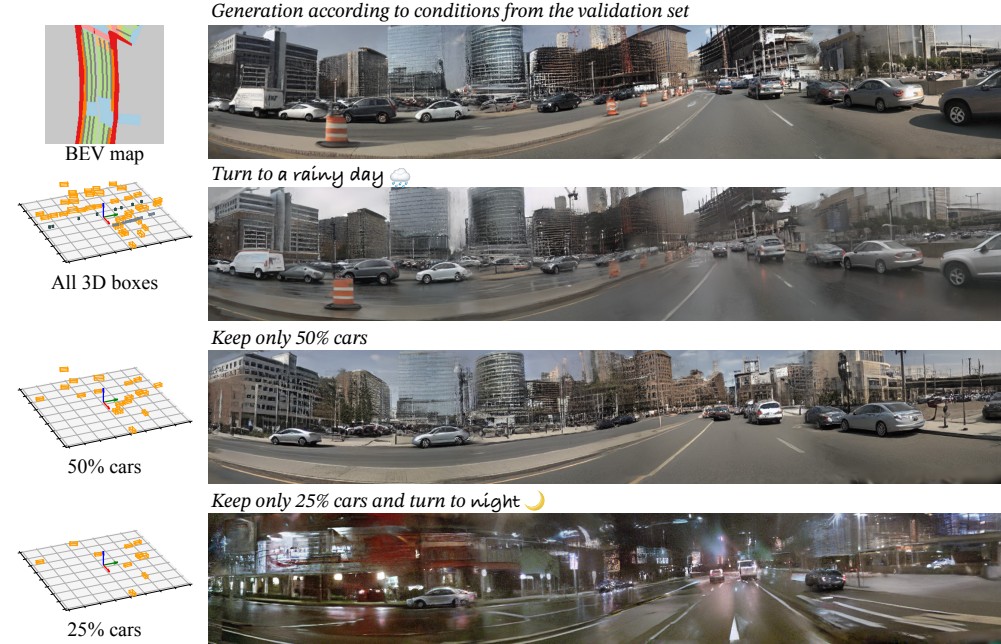

Figure 6: Qualitative evaluation for controllability (we show the view from back-left to front-right area). By changing different control signals, *MagicDrive3D* can edit the scene from different levels.

## 5.2 MAIN RESULTS

**Generation Quality**. As shown in Table 2, the evaluation of generation quality involves two aspects. Firstly, the quality of video generation is assessed using extended 16-frame MagicDrive (Gao et al., 2024) as a baseline. Despite minor improvement in single-frame quality based on FID, *Magic-Drive3D* substantially enhances video quality (as evidenced by FVD), demonstrating the efficacy of the proposed relative camera pose embedding in enhancing temporal consistency. Secondly, the image quality of renderings from the generated scene is evaluated using FID. We also include qualitative comparisons in Figure 5. Compared to 3DGS, our enhanced DGS significantly enhances visual quality in reconstructing contents, particularly in unseen novel views. More qualitative comparison can be found in Appendix C.

Table 2: Generation quality evaluation. All validation scenes from the nuScenes dataset are adopted. We use all generated views for reconstruction. Novel views adopt camera poses different from the nuScenes.

| Methods | FVD | FID (seen) | FID (novel) |
|---|---|---|---|
| Gao et al. (2024) | 177.26 | 20.92 | N/A |
| Ours (video gen.) | 164.72 | 20.67 | N/A |
| 3DGS | N/A | 45.07 | 145.72 |
| Ours (scene gen.) | N/A | 23.99 | 34.45 |

**Reconstruction Quality**. Our enhanced DGS, as a reconstruction method, is further evaluated by comparing renderings with ground truth (GT) images. Here, the generated views from the video generator are treated as GT. We employ two settings per Table 1, with results displayed in Table 3. As per all metrics, our enhanced DGS not only improves reconstruction quality for training views but also drastically enhances quality for testing views, compared to 3DGS. We include comparison with 4DGS (Wu et al., 2024) in Appendix B.

**Controllability**. *MagicDrive3D* accepts 3D bounding boxes, BEV map, and text as control signals, each of which possesses the capacity to independently manipulate the scene. To show such controllability, we edit a scene from the nuScenes validation set, as presented in Figure 6. Clearly, *MagicDrive3D* can effectively alter the generation of the scene to align with various control signals while maintaining 3D geometric consistency.

## 5.3 ABLATION STUDY

**Ablation on Enhanced Gaussian Splatting**. As detailed in Section 4.3, three enhancements - prior, modeling, and loss - have been made to 3DGS. To evaluate their efficacy, each was ablated

Table 3: Reconstruction quality evaluation. We random sample 100 scenes from the nuScenes validation set for evaluation. "cc" refers to color correction from Barron et al. (2022). Although 3DGS does not consider appearance differences, we apply "cc" to it for fair comparisons.

| Settings | | Methods | L1 ↓ | PSNR ↑ | SSIM ↑ | LPIPS ↓ |
|---|---|---|---|---|---|---|
| train view | vary-t | 3DGS | 0.0189 | 30.1191 | 0.9261 | 0.1259 |
| | | 3DGS + cc | 0.0186 | 30.2498 | 0.9253 | 0.1258 |
| | | **Ours** | **0.0167** | **32.6001** | **0.9544** | **0.0673** |
| | 360° | 3DGS | 0.0202 | 29.4943 | 0.9187 | 0.1365 |
| | | 3DGS + cc | 0.0199 | 29.6327 | 0.9178 | 0.1366 |
| | | **Ours** | **0.0174** | **32.2104** | **0.9530** | **0.0693** |
| test view | vary-t | 3DGS | 0.0890 | 17.9879 | 0.4378 | 0.4648 |
| | | 3DGS + cc | 0.0799 | 19.1387 | 0.4814 | 0.4697 |
| | | **Ours** | **0.0738** | **19.7063** | **0.5145** | **0.4115** |
| | 360° | 3DGS | 0.0910 | 17.8322 | 0.4318 | 0.4756 |
| | | 3DGS + cc | 0.0804 | 19.0773 | 0.4777 | 0.4796 |
| | | **Ours** | **0.0622** | **21.0351** | **0.5754** | **0.3207** |

Table 4: Ablation study on enhanced DGS. We adopt the same settings as in Table 3, where 100 scenes from the nuScenes validation set are adopted.

| setting | method | L1 ↓ | PSNR ↑ | SSIM ↑ | LPIPS ↓ |
|---|---|---|---|---|---|
| vary-t | 3DGS | 0.0799 | 19.1387 | 0.4814 | 0.4697 |
| | w/o AE | 0.0822 | 18.8467 | 0.4758 | 0.4452 |
| | w/o depth scale opt. | 0.0815 | 18.8885 | 0.4767 | 0.4366 |
| | w/o depth opt. | 0.1046 | 17.2776 | 0.4399 | 0.5545 |
| | w/o xyz offset + cam | 0.0798 | 19.1657 | 0.4919 | 0.4580 |
| | **Ours** | **0.0738** | **19.7063** | **0.5145** | **0.4115** |
| 360° | 3DGS | 0.0804 | 19.0773 | 0.4777 | 0.4796 |
| | w/o AE | 0.0722 | 19.7742 | 0.5114 | 0.3791 |
| | w/o depth scale opt. | 0.0736 | 19.6501 | 0.5086 | 0.3736 |
| | w/o depth opt. | 0.0995 | 17.6707 | 0.4487 | 0.5150 |
| | w/o xyz offset + cam | 0.0798 | 19.1682 | 0.4888 | 0.4663 |
| | **Ours** | **0.0622** | **21.0351** | **0.5754** | **0.3207** |

from the final algorithm, the results of which are shown in Table 4. Notations "w/o depth scale opt." and "w/o depth opt." represent the absence of GS loss optimization for $(s_{c,t}, b_{c,t})$ and use of direct output from the monocular depth model, respectively. Each component's removal lowered the method's performance, while incorrect depth sometimes performs worse than the 3DGS baseline. Removal of AE in "vary-t" led to inferior PSNR but improved LPIPS, which is reasonable because AE mitigates the pixel-wise color constraint during reconstruction.

**Ablation on Offset Choice for Deformable GS**. In addition to the overall module ablation, we observe that for Deformable GS, beyond the Gaussians' center coordinates, their attributes (including anisotropic covariance, opacity, and harmonic coefficients) can also be utilized to address local inconsistencies. To verify the effects of different choices, we randomly select 10 scenes from the nuScenes validation set for experimentation. As shown in Table 5, the results obtained by adding offsets to the center coordinates (xyz) are the best. This aligns with our observation that local inconsistencies in the generated views occur primarily in the shape of objects, and thus, xyz displacements can most effectively resolve these inconsistencies.

## 5.4 APPLICATION

**Training Support for Perception Tasks**. We demonstrate an application wherein street scene generation serves as a data engine for perception tasks, leveraging the advantage of any-view rendering to improve viewpoint robustness (Klinghoffer et al., 2023). We employ CVT (Zhou & Krähenbühl, 2022) and the BEV segmentation task following the evaluation protocols of Zhou & Krähenbühl

Table 5: Ablating comparison with offsets on anisotropic covariance (Cov.), opacity, and harmonic coefficients (Features) properties in GS. We randomly sample 10 scenes from the nuScenes validation set for experiments and apply color correction (cc) to all the renderings.

| Methods | L1 ↓ | PSNR ↑ | SSIM ↑ | LPIPS ↓ |
|---|---|---|---|---|
| 3DGS | 0.0733 | 19.7514 | 0.5210 | 0.4496 |
| Features offset | 0.0624 | 20.9882 | 0.5940 | 0.3463 |
| Cov. offset | 0.0632 | 20.8133 | 0.5854 | 0.3626 |
| Opacity offset | 0.0656 | 20.5332 | 0.5733 | 0.3845 |
| **Ours (xyz offset)** | **0.0546** | **21.9428** | **0.6288** | **0.2759** |

Table 6: *MagicDrive3D* improves the viewpoint robustness (Klinghoffer et al., 2023) of CVT (Zhou & Krähenbühl, 2022). All results are mIoU for BEV segmentation. Colors highlight the differences with baseline. The best results are in **bold**.

| Setting | Method | no rig | depth+0.5m | pitch-5° | yaw+5° | yaw-5° |
|---|---|---|---|---|---|---|
| vehicle | only real data | 17.14 | 16.63 | 15.50 | 16.99 | 15.94 |
| | w/ render view (no rig) | 20.67 +3.53 | 20.13 +3.50 | 17.03 +1.53 | 19.40 +2.41 | 19.30 +3.36 |
| | w/ random aug. of 4 rigs | **21.05** +3.91 | **20.46** +3.83 | **19.75** +4.25 | **19.81** +2.82 | **19.83** +3.89 |
| road | only real data | 54.94 | 54.56 | 53.82 | 54.20 | 53.67 |
| | w/ render view (no rig) | 60.31 +5.37 | 59.93 +5.37 | 58.46 +4.64 | 59.16 +4.96 | 59.32 +5.65 |
| | w/ random aug. of 4 rigs | **60.59** +5.65 | **60.38** +5.82 | **59.95** +6.13 | **60.21** +6.01 | **60.29** +6.62 |

(2022); Gao et al. (2024). By incorporating 4 different rigs on the FRONT camera and adding rendered views for training, the negative impact from viewpoint changes is alleviated (Table 6), exemplifying the utility of street scene generation in training perception tasks.

**Render Object-level Dynamic**. *MagicDrive3D* generates 3DGS representations of scenes, thereby enabling applications for scene editing. Our approach employs metric scale modeling, ensuring that the editing of scenes corresponds accurately to real-world physical distances. As demonstrated in Figure 7, we segmented the generated GS and relocated the object on the right. The resulting scene GS supports rendering effectively.

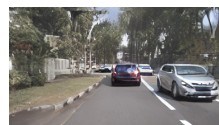 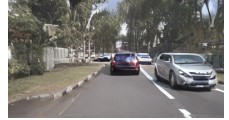 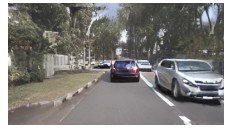 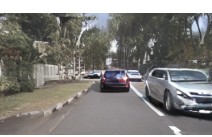

(a) Original view.    (b) Move forward 0.1m.    (c) Move forward 1m.    (d) Move forward 1.6m.

Figure 7: Application on rendering object-level dynamic. After scene generation, we can segment and move the vehicle (the one on the right) in 3D to render a dynamic object.

## 6    CONCLUSION AND DISCUSSION

This paper introduces *MagicDrive3D*, a unique 3D street scene generation framework that integrates geometry-free view synthesis and geometry-focused 3D representations. *MagicDrive3D* significantly reduces data requirements, enabling training on typical autonomous driving datasets, such as nuScenes. Within the generation-reconstruction pipeline, *MagicDrive3D* employs a video generation model to enhance inter-frame consistency, while the enhanced deformable GS improves reconstruction quality from generated views. Comprehensive experiments demonstrate that *Magic-Drive3D* can produce high-quality 3D street scenes with multi-level controls. Additionally, we show that scene generation can serve as a data engine for perception tasks such as BEV segmentation.

**Limitation and Future Work**. As a data-centric method, *MagicDrive3D* sometimes struggles to generate complex objects like pedestrians, whose appearances are intricate. Additionally, areas with much texture detail (e.g., road fences) or small spatial features (e.g., light poles) are occasionally poorly generated due to limitations in the reconstruction method. Future work may focus on addressing these challenges and further improving the quality and robustness of generated 3D scenes.

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

# APPENDIX

## A   MORE IMPLEMENTATION DETAILS

For the monocular depth model, we use ZoeDepth (Bhat et al., 2023). Although it is trained for metric depth estimation, due to domain differences, raw estimation is not usable, as shown in Section 5.3 and Figure 3. Methodologically, *MagicDrive3D* does not rely on a specific depth estimation model. Better estimations can further improve our scene generation quality.

Since GS only supports perspective rendering, to stitch the view for panorama, we use code from perspective to equirectangular transformation provided by https://github.com/timy90022/Perspective-and-Equirectangular.

All our experiments are conducted with NVIDIA V100 32GB GPUs. The generation of a single scene takes about 2 minutes for video generation and about 30 minutes for deformable GS reconstruction. For reference, 3DGS reconstruction typically takes about 23 minutes for scenes of similar scales. Therefore, the proposed enhancement is efficient. As for rendering, there is no additional computation for our method compared with 3DGS.

## B   MORE RECONSTRUCTION BASELINE

Focusing on dynamic scenes, 4DGS (Wu et al., 2024) introduces comprehensive improvements over 3DGS and achieves notable results. Therefore, we replace deformable GS with 4DGS. As shown in Table 7, by incorporating non-rigid dynamics, 4DGS already performs better than 3DGS. However, in our task, 4DGS underperforms compared to our deformable GS. Based on the results in Table 5, we hypothesize that our reconstruction algorithm only needs to address local dynamics caused by content inconsistency. Allocating excessive dynamics at the representational level may hinder model convergence, thus 4DGS does not yield better results on our scenarios.

Table 7: Comparison with 4DGS (Wu et al., 2024). We randomly sample 10 scenes from the nuScenes validation set for experiments and apply color correction (cc) to all the renderings.

| Methods | L1 ↓ | PSNR ↑ | SSIM ↑ | LPIPS ↓ |
|---|---|---|---|---|
| 3DGS | 0.0733 | 19.7514 | 0.5210 | 0.4496 |
| 4DGS | 0.0601 | 21.1195 | 0.5892 | 0.4475 |
| **Ours** | **0.0546** | **21.9428** | **0.6288** | **0.2759** |

## C   COMPARISON WITH SIMPLE BASELINES

As shown in Figure 8, we further compare *MagicDrive3D* with two baselines, *i.e.*, Lucid-Dreamer (Chung et al., 2023) and WonderJourney (Yu et al., 2024). The former method has been proposed recently and takes text description as the only condition. Thus, it is hard to generate photo-realistic street scenes. When providing multi-view video frames from nuScenes with known camera poses, their pipeline fails to reconstruct. We suppose the reason is limited overlaps and errors from depth estimation. As suggested by the released code, we changed the image generation model to `lllyasviel/control_v11p_sd15_inpaint` for inpainting by providing a nuScenes image, *i.e.*, Figure 8a. However, due to the lack of controllability, the results from LucidDream (*e.g.*, Figure 8b) are unsatisfactory. On the other hand, due to the lack of control over objects within the scene, WonderJourney struggles to generate coherent scenes. Inpainting-based methods like the two above exhibit a pronounced sense of patches and face significant challenges in achieving 360° coverage.

Figure 8d further shows directly stitching real data. It is also bad due to the limited overlaps between views. On the contrary, the scene generated from *MagicDrive3D* can render continuous panorama, as shown in Figure 8e, which is also controllable through multiple conditions.

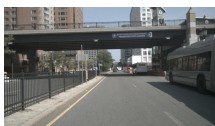 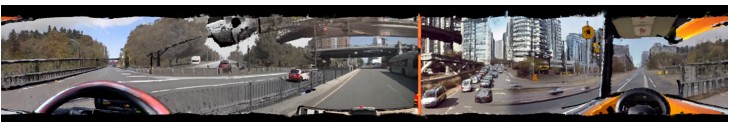

(a) Conditional image to LucidDreamer (Chung et al., 2023)

(b) Scene generated by LucidDreamer (Chung et al., 2023), with text "*A driving scene in the city from the front camera of the vehicle. A bus on the right side. There is a bridge overhead. There is a railing in the center of the left road. Some vehicles ahead*"

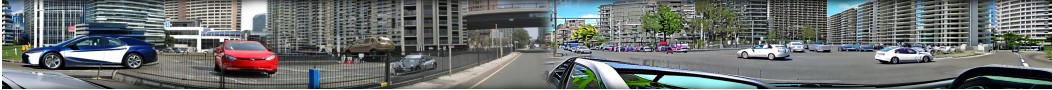

(c) Scene generated by WonderJourney (Yu et al., 2024). We set the focal length to be the same as the conditional image. WonderJourney cannot control road semantics (many objects are physically implausible) and fails in loop closure for 360° scene generation. Conditions are the same as Figure 8b.

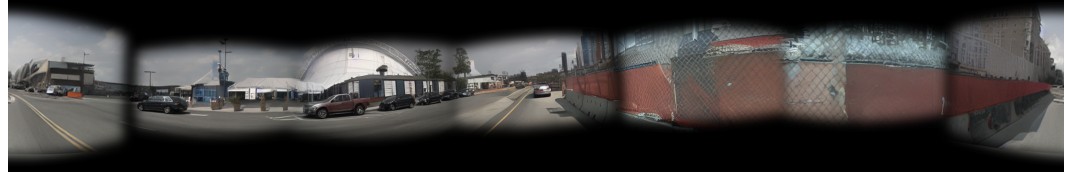

(d) Stitched panorama with real camera views from nuScenes dataset. Due to the limited overlaps, there are many empty (black) areas.

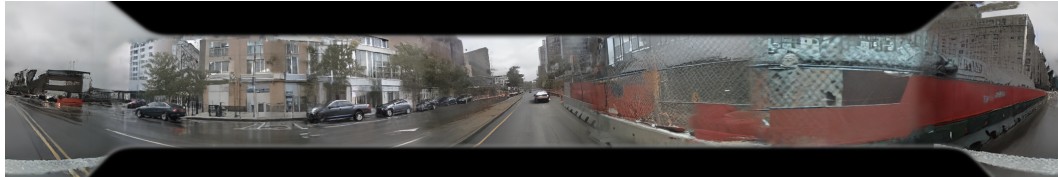

(e) Panorama from *MagicDrive3D*. The scene is generated with the same object boxes and BEV map as Figure 8d, but turned to "*rainy day*".

Figure 8: Comparison with two baselines (LucidDreamer (Chung et al., 2023) and WonderJourney (Yu et al., 2024)) and direct stitching real images.

Note that, panorama generation is only one of the applications of our generated scenes. We show them just for convenient qualitative comparison within the paper. Since our scene generation contains geometric information, they can be rendered from any camera view, as shown in Figure 5.

## D IMPLEMENTATION DETAIL OF APPEARANCE EMBEDDING

We show in Figure 9 the detailed architecture of the CNN used in our appearance modeling. The AE map is 32× smaller than the input image to reduce the computational cost. Hence, we first downsample the input image by 32×. Then, we use 3 × 3 convolution for feature extraction and pixel shuffle for upsampling. Each convolution layer is activated by ReLU.

## E BROADER IMPACTS

The implementation of *MagicDrive3D* in controllable 3D street scene generation could potentially revolutionize the autonomous driving industry. By creating detailed 3D scenarios, self-driving vehicles can be trained more effectively and efficiently for real-world applications, thereby leading to improved safety and accuracy. Moreover, it could potentially provide realistic simulations for human-operated vehicle testing and training, thus contributing to reducing the occurrence of accidents on the roads while enhancing driver expertise. In the broader scope, *MagicDrive3D* could be of considerable value to the virtual reality industry and video gaming industry, enabling these sectors to generate more lifelike 3D scenes and intricate gaming experiences.

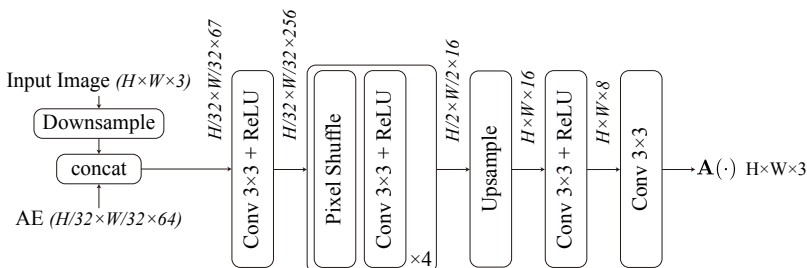

Figure 9: The CNN architecture of appearance modeling, as introduced in Section 4.3.

On the downside, the development and application of such advanced technology could lead to certain unwanted scenarios. For instance, the increased automation in industries, driven by the potential of this technology, could lead to job losses for drivers and other related professionals as their roles become automated. A societal transition will be needed to avoid negative impacts on employment levels and the fairness of wealth distribution.

## F    MORE QUALITATIVE RESULTS

We show more generated street scenes from *MagicDrive3D* in Figure 10 and Figure 11.

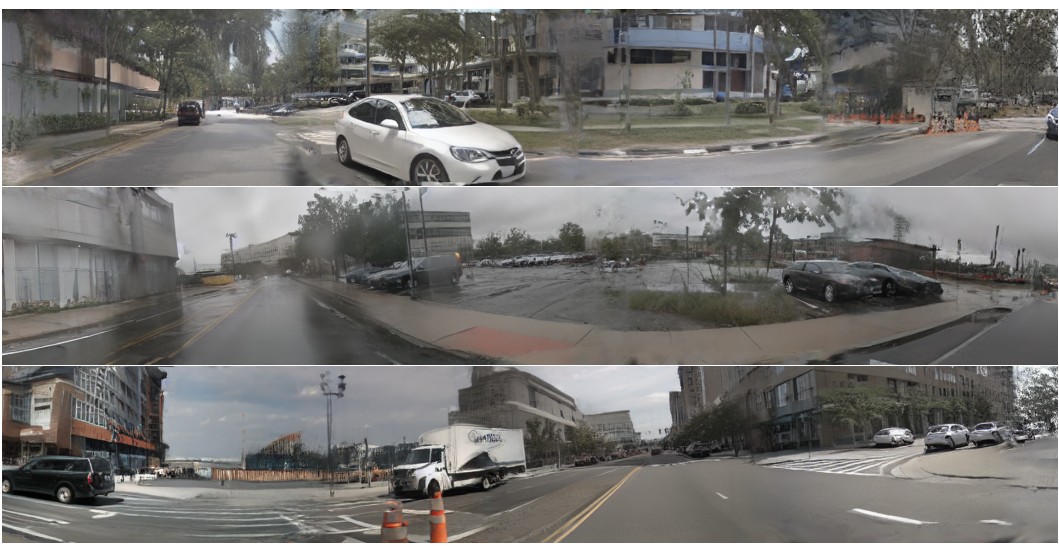

Figure 10: Generated street scenes from *MagicDrive3D*. We adopt control signals from nuScenes validation set. We crop the center part for better visualization.

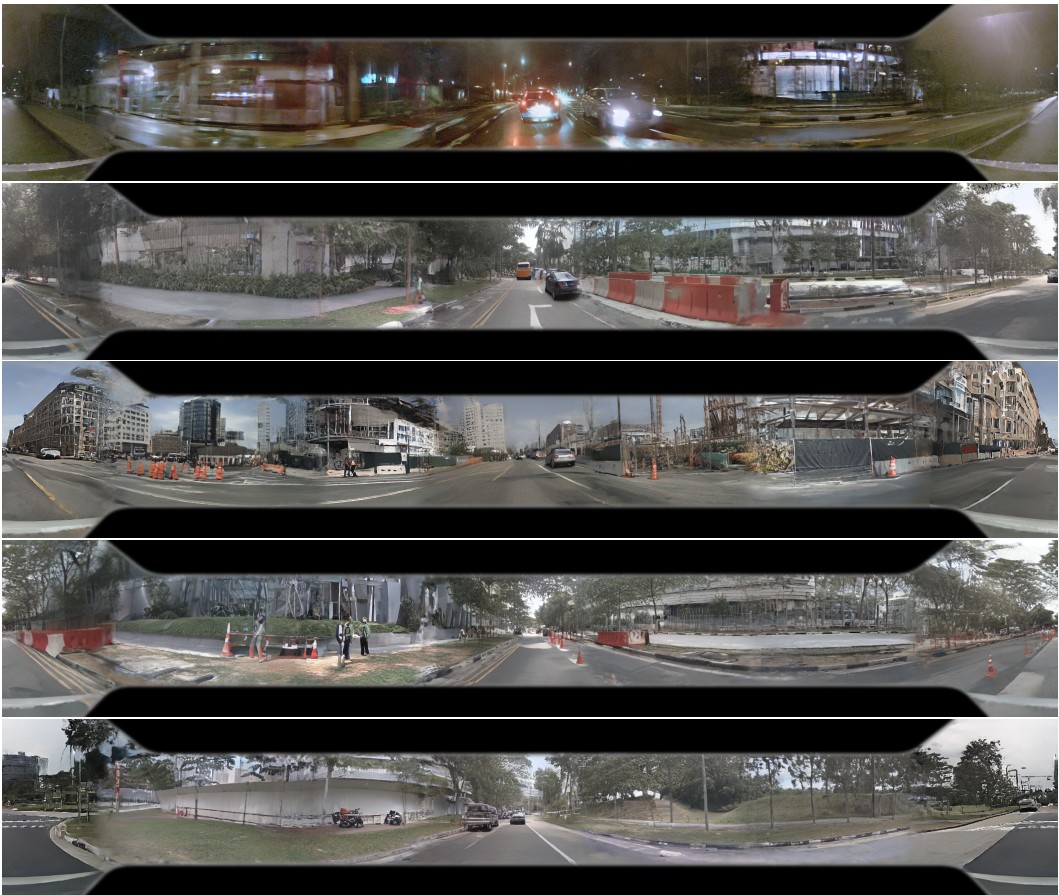

Figure 11: Generated street scenes from *MagicDrive3D*. We adopt control signals from nuScenes validation set. The black regions are not fully covered, constrained by the camera's FOV.

