# OpenReview forum: "MagicDrive3D: Controllable 3D Generation for Any-View Rendering in Street Scenes"
_ICLR.cc/2025/Conference — ICLR 2025 Conference Withdrawn Submission_

### Official Review · Reviewer_s5XG · 2024-10-15

**Soundness:** 3
**Presentation:** 3
**Contribution:** 2
**Rating:** 5
**Confidence:** 4

**Summary:**

The paper presents MagicDrive3D, a novel framework for controllable 3D street scene generation. The framework combines geometry-free video generation with geometry-focused reconstruction using 3DGS. MagicDrive3D allows for multi-condition control including BEV maps, 3D objects, and text descriptions, enabling the generation of diverse and high-quality 3D street scenes. It also improves downstream tasks like BEV segmentation and supports realistic scene simulations for applications such as autonomous driving.

**Strengths:**

1. The proposed framework supports controllable scene generation using BEV maps, 3D bounding boxes, and text descriptions, which enhances its applicability in tasks like autonomous driving simulations.
2. The introduction of deformable 3D GS effectively addresses local dynamics and exposure discrepancies, ensuring better scene generation quality.

**Weaknesses:**

1. MagicDrive3D is composed of two parts: a video generation model and a 3DGS to recover 3D scenes from images, both are proposed in previous works, while showing technical improvements, still limiting the overall novelty of the paper.
2. The comparison in Table 2 is only made with Vallinia 3D-GS, yet there are several other dynamic 3D-GS methods for road scenes (e.g., PVG[1], StreetGaussian[2]) that should also be considered for comparison.

[1] Periodic Vibration Gaussian: Dynamic Urban Scene Reconstruction and Real-time Rendering
[2] Street Gaussians: Modeling Dynamic Urban Scenes with Gaussian Splatting

**Questions:**

I'm a little questioned about the quality of the generated night scene in Figure 1, as it's blurry and doesn’t clearly convey a night setting.

---

### Official Review · Reviewer_vviZ · 2024-10-28

**Soundness:** 3
**Presentation:** 3
**Contribution:** 3
**Rating:** 6
**Confidence:** 5

**Summary:**

The manuscript introduces MagicDrive3D, a novel approach for controllable 3D street scene generation. This method divides the generation process into two distinct stages. In the first stage, a conditional generation model is trained to produce multi-view video sequences from the perspective of an ego car. The authors enhance the existing MagicDrive framework by encoding the relative pose with respect to the first frame and using these encodings as conditions for the network. In the second stage, the focus shifts to reconstruction, where the generated data is used to reconstruct the 3D scene. The authors propose several improvements to the 3DGS in terms of spatial location priors, modeling, and loss functions, specifically tailored for street view scene reconstruction. Experimental results demonstrate the effectiveness of each proposed component.

**Strengths:**

1. The paper is well-structured and straightforward to understand.
2. The concept of breaking down 3D scene generation into a sequential multi-view generative stage followed by a static reconstruction stage, utilizing two distinct representations that have proven effective in their respective areas, is particularly intriguing.
3. The ablation studies demonstrate a significant improvement over the selected baselines (3DGS and LucidDreamer).

**Weaknesses:**

1. The performance on test views is not particularly strong. As noted in the manuscript, the PSNR on novel views in both test settings is below 22. While this work does advance the field of scene generation, it is not yet suitable for practical applications, such as generating synthetic data for end-to-end autonomous driving policy training.
2. The manuscript lacks a comparison with key baselines during the reconstruction phase, specifically Street Gaussians [A].
3. Have you attempted a long-term rollout of video diffusion models? If such a long-term rollout were conducted (like Vista [B]), would the two-stage scene generation pipeline still perform effectively?


[A] Street Gaussians: Modeling Dynamic Urban Scenes with Gaussian Splatting
[B] Vista: A Generalizable Driving World Model with High Fidelity and Versatile Controllability

**Questions:**

Please see the section of weaknesses.

---

### Official Review · Reviewer_wYwF · 2024-11-02

**Soundness:** 2
**Presentation:** 2
**Contribution:** 2
**Rating:** 3
**Confidence:** 5

**Summary:**

This paper introduces MagicDrive3D, a new framework for controllable 3D street scene generation useful for view synthesis. The framework supports multi-condition control, including BEV road maps, 3D object bounding boxes, and text descriptions. The proposed framework MagicDrive3D first trains a video generation model and then reconstructs from the generated data.

**Strengths:**

- [S1: Significance] The paper addresses an important problem in the field of computer vision: controllable 3D scene generation. The proposed method has the potential to be used in a variety of applications, including autonomous driving simulation, virtual reality, and video gaming.

**Weaknesses:**

- [W1] The technical contributions of pose conditioned video generation and its relation in the framework is not clearly stated.
  - [W1.1] According to Figure 2, it looks like the video generator works without conditioning on input camera images. If that is the case, the reviewer would like to understand what’s the benefit of feeding the video generated multi-view data to Stage 2 compared to using ground-truth data? Based on my understanding, the exposure discrepancy across multi-views and dynamic objects in the generated data will pose the same challenge to Stage 2 (vs. ground-truth camera images).
  - [W1.2] If the proposed video generator works without conditioning on camera input images, please explain the steps that generate row (e) in Figure 8. In Figure 8, it is clear that the proposed system is able to take camera images as input and apply style transfer on top.
  - [W1.3] The reviewer cannot find any videos in supplementary material, which is usually the hard requirement for accepting a video generation paper. The reviewer feels video results are still required for this paper, as it highlights video generation as one important step compared to existing work in 3D street view generation.

- [W2] The paper’s claim that Magic3D is the first to achieve controllable 3D street scene generation using a common driving dataset (Line 91-92) is questionable. For example, controllable 3D street scene generation has been achieved in Panoptic Neural Fields [NewRef1] on KITTI dataset. In another example, as discussed in Section 5.1 of BlockNeRF [NewRef2], 3D street scene generation has also been achieved on the single-capture subset (open-sourced) called San Francisco Mission Bay Dataset. Please discuss the relevant work in the main text and compare against them for novel view synthesis (show quantitative metrics).
  - [W2.1] The reviewer would recommend to conduct a more sophisticated literature review. For example, this paper also missed prior work that shares similar motivation but on object reconstruction from driving videos using a generative model GINA-3D [NewRef3].

- [W3] Important details regarding the FVD and FID metrics are missing. As Nuscenes dataset is relatively small, the reviewer would like to understand how many images or 16-frame video clips have been used in computing the metrics. How do you construct the real videos and generated videos (on what conditions). This is an important factor to decide whether the metrics reported in Table 2 are valid or not.
  - [W3.1] In the field of image and video generation, it is known that FID and FVD are good but not perfect. Certain adversarial artifacts can lead to unexpected changes to FID and FVD. Please consider using FID-DINOv2 [NewRef4] and FVD-VideoMAEv2 [NewRef5] as alternative metrics.

- [W4] While one focus of the paper is on controllable generation, the reviewer cannot find enough details on different controllable signals. It would be good to develop quantitative metrics to measure the accuracy of control and provide more diverse examples of scene editing. This could include user studies to assess the usability and effectiveness of the control mechanisms.

- [W5] The paper focuses on 3D street view synthesis but the reviewer cannot find 3D visualizations in the supplementary materials.


References
- [NewRef1] Panoptic Neural Fields: A Semantic Object-Aware Neural Scene Representation, Kundu et al., In CVPR 2022.
- [NewRef2] Block-NeRF: Scalable Large Scene Neural View Synthesis, Tancik et al., In CVPR 2022.
- [NewRef3] GINA-3D: Learning to Generate Implicit Neural Assets in the Wild, Shen et al., In CVPR 2023.
- [NewRef4] Exposing flaws of generative model evaluation metrics and their unfair treatment of diffusion models, Stein et al., In NeurIPS’23.
- [NewRef5] On the Content Bias in Fréchet Video Distance, Ge et al., In CVPR 2024.

**Questions:**

Please address the concerns raised in the weaknesses section.

---

### Official Review · Reviewer_T2U1 · 2024-11-06

**Soundness:** 3
**Presentation:** 3
**Contribution:** 3
**Rating:** 6
**Confidence:** 5

**Summary:**

This paper proposes a novel approach for 3D street scene generation, with a strong emphasis on multi-condition controllability, including BEV (Bird’s Eye View) maps, 3D objects, and text descriptions. The method involves first training a video generation model, followed by scene reconstruction using deformable Gaussian splatting (DGS). This two-step approach improves the quality and temporal consistency of the generated scenes, making it particularly beneficial for data augmentation in downstream tasks. Validation on the nuScenes dataset highlights the method’s strengths in both controllability and scene reconstruction quality.

**Strengths:**

The paper presents impressive visualization results, with generated scenes that are virtually indistinguishable from real-world counterparts.

It introduces an innovative generation-first, reconstruction-later pipeline, which simplifies both scene control and data acquisition, offering a more streamlined approach to 3D scene synthesis.

The deformable Gaussian splatting (DGS) method significantly enhances the quality of both generated and reconstructed views, demonstrating robust performance in complex autonomous driving environments.

The method provides high controllability through multi-level signals, including BEV maps, 3D bounding boxes, and text descriptions, enabling precise and flexible scene generation.

**Weaknesses:**

The method occasionally struggles with generating intricate objects, such as pedestrians, and detailed texture areas, like road fences, which can affect the realism of the scenes in certain contexts.

The experiments are conducted solely on the nuScenes dataset, which includes 700 training and 150 validation clips. Although widely used, this dataset may not fully capture the complexity of real-world environments, raising concerns about the method’s generalizability to more diverse and challenging scenarios.

The scholarship could be improved by referencing recent advancements in street-view generation, such as SCP-Diff: Photo-Realistic Semantic Image Synthesis with Spatial-Categorical Joint Prior [ECCV 2024]. This would help position the proposed approach more clearly within the current state of the field.

**Questions:**

see weakness box.

---

### Note · Authors · 2024-11-14

I have read and agree with the venue's withdrawal policy on behalf of myself and my co-authors.